# Effect of Ultrasonic-Assisted Modification Treatment on the Microstructure and Properties of A356 Alloy

**DOI:** 10.3390/ma15103714

**Published:** 2022-05-22

**Authors:** Xinyi Hu, Dongfu Song, Huiping Wang, Yiwang Jia, Haiping Zou, Mingjuan Chen

**Affiliations:** 1School of Mathematics, Physics and Statistics, Shanghai University of Engineering Science, Shanghai 201620, China; hxy130119114@163.com; 2Institute of New Materials Guangdong Academy of Sciences, Guangzhou 510650, China; ywjia2018@163.com; 3Jiangxi Qian Yue New Materials Co., Ltd., Ganzhou 341000, China; zhpylcip@126.com (H.Z.); chenmingjuan10@163.com (M.C.)

**Keywords:** ultrasonic treatment, Sr modification, Sr/Ce synergistic modification, eutectic Si

## Abstract

Ultrasonic treatment was applied to an A356 aluminum melt with different modifiers, and the effects of ultrasonic treatment on the structure and properties of the A356 alloy were studied. The results showed that α-Al was effectively refined with different ultrasonic modification treatments. In particular, ultrasonic treatment showed the most obvious refinement with macroscopic grains of unmodified alloy and optimized the refinement of secondary dendrite arm spacings in the Sr/Ce synergistic alloys. The eutectic Si of the unmodified A356 alloy had no obvious change after the ultrasonic treatment, but the branch diameter of the eutectic Si reduced in the Sr and Sr/Ce modification alloys after the ultrasonic treatment. The ultrasonic treatment significantly improved the ultimate tensile strength and elongation of the as-cast A356 alloy with the unmodified material, which was due to refinement of the α-Al grains by the ultrasonic treatment. After the T6 heat treatment, the ultimate tensile strength values of the alloys showed no obvious change due to the ultrasonic treatment, but the plasticity of the alloy was significantly improved. Mg_2_Si precipitation was the dominant strengthening mechanism during the T6 heat treatment, while the plasticity was determined by the size and distribution of the eutectic Si. Acoustic cavitation caused by the ultrasound-activated impurities and the induced heterogeneous nucleation and supercooled nucleation in the groove melt was the main cause of the α-Al refinement, the eutectic Si modification and the improvement in the mechanical properties.

## 1. Introduction

The A356 alloy is widely used in lightweight automotive materials, electronic communications and other applications, owing to its excellent casting properties, corrosion resistance and high toughness [1,2,3]. Limited by the molding processes of Al alloy castings, such as low-pressure casting, gravity casting and counterpressure casting [4,5], the melt solidifies at a lower cooling rate, which results in a coarser organization of the alloy. In particular, eutectic Si with lamellae severely splits the matrix [6], which causes stress concentration and greatly reduces the toughness of the alloy.

Structural refinement is an effective method for improving the toughness of the castings [7]. In recent decades, scholars [8,9,10,11,12,13,14,15] have carried out extensive research on the refinement of the matrix and second phase and realized many beneficial results. Elements such as Na, Sr, Sb, P and rare earth elements (REs) are commonly used for eutectic refinement and are also referred to as modifiers. In particular, Sr is an excellent modifier due to its long-term modification effect, which is applied in many fields. However, Sr has almost no effect on the refinement of the matrix structure (α-Al) [1,16], which precludes further improvements in the toughness of the Al castings.

In recent years, researchers have tried to refine and modify A356 alloys with synergistic modifiers. Qiu et al. reported that Al-6Sr-7La reduced the secondary dendrite arm spacing (SDAS) of the matrix and also modified the morphology of eutectic Si from plates to fibrous and granular. The ultimate tensile strength (UTS), yield strength (YS), elongation (EL) and hardness of the alloy were increased by 13.8%, 7.7%, 52.4% and 6.4%, respectively, with addition of 0.5 wt.% Al-6Sr-7La [16]. Wu added the Al-5Sr-8Ce master alloy to an A356 alloy and found that the addition of Sr/Ce refined the grain size of the A356 alloy and decreased the SDAS [1]. However, RE elements are expensive in the actual production of Al alloys. It was found that large amounts of added RE tend to be enriched, resulting in coarse rare earth-containing intermetallic compounds (Re-IMCs) [17,18]. These coarse Re-IMCs are susceptible to becoming the cores of crack extensions due to severe stress concentrations and damage to the mechanical performance of Al alloys.

Based on the cavitation and acoustic flow effects of ultrasonication, ultrasonic treatment accelerates melt flow by the periodic growth and explosion of air bubbles, thus changing the pressure and temperature field of the melt [19,20,21,22,23]. In addition, ultrasonic treatment has many advantages; it is a simple process, it exhibits low cost and it has no significant impact on the melt composition [24]. Zhao reported that the average grain sizes of as-cast 7050 alloys were reduced by more than 50% with ultrasonic treatment [25]. Zhang et al. prepared AlSi7Mg-1.0Ce semisolid ingots with the ultrasonic treatment, and the average grain diameter and shape factor of the primary α-Al were reduced to 57.54 μm and 1.33, respectively [26]. Peng et al. indicated that the segregation index of Cu elements in the semicontinuous casting Al-Cu alloys was reduced from 0.23 to 0.15 after ultrasonic treatment [27]. Qi et al. revealed that ultrasonic treatment refined the primary α-Al grains and also reduced the length of the β-Al_5_FeSi phase in the Al-8Si-0.55Fe alloy from 18 μm to 12 μm, and the corresponding uniformity of the distribution was improved [28]. Puga et al. reported that the average grain diameter of α-Al in an AlSi9Cu3 alloy was reduced to less than 50 μm by ultrasonic treatment, and the average area of eutectic Si decreased to less than 10 μm^2^; however, the mechanism of modification was not clear [24]. Research on ultrasonic treatment is currently focused primarily on grains, composition and intermetallic compounds, and there is a lack of in-depth research on the morphology and distribution of eutectic Si.

In this study, A356 alloys with different modifiers were subjected to ultrasound. The effects of ultrasonic treatment on the microstructure and mechanical properties of A356 alloys were studied, and the strengthening mechanism is discussed in detail.

## 2. Materials and Methods

### 2.1. Specimens Fabrication

The experimental materials consisted of commercial pure Al, Mg, Al-20Si, Al-5Sr and Al-5Sr-5Ce master alloys. The preparation process was as follows: first, approximately 2 kg of raw material (commercial pure Al and Al-20Si) was melted at 750 °C, held for 30 min and then cooled to 700–720 °C by furnace cooling. Second, Mg and modifier were added to the melt by using titanium tools, approximately 10 g of refining agent was added to the melt after 5–15 min and the melt was stirred slowly with titanium tools. Third, the melt was held for 30 min, and then the slag was removed. In the fourth step, the acoustic radiator of the ultrasonic vibration equipment (HC-2000E-QC, 1500 W, 20 kHz) was preheated for 2 min and inserted into the melt to a depth of approximately 1 cm (as shown in Figure 1), and the ultrasonic treatment time was 120 s. The experiment process was protected by 99.99% argon. Finally, the slag was removed and cast into a rectangle-shaped mold (200 × 30 × 120 mm^3^) preheated to 260 °C, and an ingot was obtained by air cooling.

### 2.2. Microstructure Characterization and Properties Test

Samples were cut by wire-cut electrical discharge machining, the compositions of alloys were analyzed by a spectrum analyzer (SPECTRO-MAX, SPECTRO Analytical Instruments, Kleve, Germany), and the results are listed in Table 1. The T6 treatment process included solid solution aging, holding for 4 h at 535 ± 5 °C first and quenching in hot water in a range of 90–100 °C. After 12 h, these samples were heated to 175 °C and held for 5 h. Finally, the samples were cooled by air. The metallographic samples were prepared by grinding with 40–2000 sandpaper, polishing with 1% MgO aqueous solution and etching in 0.5% HF aqueous solution for 10 s. To obtain the macroscopic grains of the alloy, the samples were etched in an aqueous solution of 60 vol% HCl, 30 vol% HNO_3_ and 5 vol% HF for 30 s after being ground. The three-dimensional morphology of eutectic Si was obtained by etching in 20% NaOH aqueous solution for 0.5–2 h. Then, the corroded surface of the sample was cleaned with 10% HNO_3_, repeatedly washed with pure water and then dried with cold air.

The microstructures and phase compositions of the alloys were analyzed by optical microscopy (LEICA DMI3000 M), scanning electron microscopy (SEM, Gemini 300, ZEISS, Jena, Germany) and energy dispersive spectrometry (EDS, OXFORD-7412, Oxford, UK). Image Pro-Plus 6.0 software was used to analyze the morphological characteristics of the microstructures, including the SDAS, area and aspect ratio of the second phase. The statistics data of the originated from 30 metallographic photographs. The multiple of images used to count the morphological characteristics of α-Al and the eutectic silicon were 50 and 1000, respectively, and the total analysis area were about 90 mm^2^ and 2.25 mm^2^, respectively. Tensile tests (GB/T228.1-2010) were carried out at room temperature by a material testing machine (DNS200, Sinotest Equiment Co., Ltd., Changchun, China) with a strain rate of 2 mm/min. The average values of the three groups were taken as the tensile properties of the alloy, and the morphology of tensile fracture was observed by scanning electron microscopy. The solidification curve of the alloy was measured by power-compensated DSC equipment (STA409PC, NETZSCH, Selb, Germany)

## 3. Experimental Results

### 3.1. Effect of Ultrasonic Treatment on the Microstructure of the A356 Alloy

The effects of the different treatment conditions on the structure of the as-cast A356 alloy are shown in Figure 2. Figure 2a–c shows that the macrostructure of the alloy without ultrasonic treatment comprised large and irregular grains. Without a modifier, the microstructure consisted of fine columnar crystals at the edges and large equiaxed crystals at the center. With the addition of the modifier, the grain size was significantly coarsened, long columnar crystals appeared at the edges and the maximum size of the equiaxed grains in the central area was more than 5 mm. In particular, grain coarsening by Sr/Ce synergistic modification was more serious than that for Sr modification. Figure 2d–f shows that the grains of the alloys with different modifiers added were significantly refined and homogenized by ultrasonic treatment, and most of the grains were less than 1 mm in diameter. However, there were still a few coarse equiaxed grains in the central areas of the alloys with Sr modification. Therefore, the ultrasonic treatment had the best refinement effect on the unmodified alloys, followed by Sr/Ce synergistic modification and Sr modification.

Figure 3a–c shows the morphologies of α-Al without the ultrasonic treatment; they exhibit well-developed dendrites, and there were no significant differences among morphologies resulting from the different modification methods. After the ultrasonic treatment (Figure 3d–f), the α-Al dendrites in the A356U alloy were transformed into petal-like shapes, as were those of the A356SU and A356SCU alloys. In addition, the SDAS values of the alloys with different modifiers were significantly reduced.

Figure 4 shows that the SDAS values of the A356 alloys were mainly distributed between 29–42 μm and showed an average of 37 μm, which is 8% larger than that of A356U. The SDAS averages of the A356SU and A356SRU alloys were 29.7 μm and 24.1 μm, respectively, which were 8.5% and 4.2% lower than those seen without ultrasonic treatment, respectively.

Figure 5 shows the two-dimensional morphologies of the eutectic Si in the as-cast and T6 state alloys with different modifiers and ultrasonic treatments. As seen from Figure 5a,b, the eutectic Si presents an interconnected structure with coarse slats in the as-cast A356 alloy. After the T6 heat treatment of the A356 alloy without modification, the morphologies of the eutectic Si phases were transformed into independently distributed particles and short rods, which significantly reduced the connectivity of the eutectic Si phases. Figure 5d shows that the size of the eutectic Si was slightly reduced by the ultrasonic treatment. Figure 5e,g,i,k show that the morphology of the eutectic Si exhibited fiber clusters in the as-cast states of the A356S and A356SC alloys. After the T6 heat treatment of the modified A356, the morphology of the eutectic Si, which was independently distributed in the dendrite gap, adopted a regular spherical shape. It is worth noting that the ultrasonic treatment had no obvious effect on the morphology of the eutectic Si after the modifications with Sr and Sr/Ce.

Figure 6a,c shows that the deep-etched morphology of the eutectic Si in the unmodified A356 alloy, which presented a stepped structure. The microstructure consisted of two or more sheets of lath-shaped eutectic Si. This structure showed no obvious difference after the ultrasonic treatment. After the T6 heat treatment, the stepped structure mentioned above was not significantly changed, but the lath-shaped side edges and corners were spheroidized, and part of the eutectic Si was fused and transformed into fine short rods and particles (Figure 6b,d). Figure 6e,i shows that the deep-etched morphologies of the eutectic Si in the A356S and A356SC alloys, which were composed of fine skeletal and fibrous materials, and parts of the fibrous eutectic Si, had several slender branches. After the ultrasonic treatment, the branch diameter of the eutectic Si reduced obviously (Figure 6g,k). After the T6 treatment, the slender eutectic Si melted and spheroidized, as shown in Figure 6f,h,i,j. After the ultrasonic treatment, the amount of the eutectic Si incompletely spheroidized was slightly reduced.

To improve the accuracy of the measurement, the T6 heat treatment was implemented before characterizing the morphology of the eutectic Si, and the statistical results are shown in Figure 7. Figure 7a shows that the average area of the eutectic Si in A356U was approximately 27 μm^2^, which was 7% lower than the area seen without the ultrasonic treatment. The average areas of the eutectic Si in the A356SU and A356SCU alloys were approximately 6.34 μm^2^ and 6.28 μm^2^, which were 15% and 3% lower than those seen without the ultrasonic treatment, respectively. Figure 7b shows that the average aspect ratio of the eutectic Si in the A356U alloy was approximately 15.2, which was 21% lower than that seen without the ultrasonic treatment. The average aspect ratios of the A356SU and A356SCU alloys were approximately 2.03 and 1.86, respectively, which were reduced by 11.7% and 13.4% relative to those observed without the ultrasonic treatment. Therefore, the ultrasonic treatment refined the size of the eutectic Si to a certain extent and significantly improved the shape factor of the eutectic Si with different modifiers.

### 3.2. Effect of Ultrasonic Treatment on Mechanical Properties

The tensile results for the samples in the as-cast and T6 states are shown in Figure 8. Figure 8a,b shows that the UTS and EL of the alloys under different modification conditions were improved after the ultrasonic treatment. In particular, the UTS and EL of the as-cast A356U alloy reached 163 MPa and 4.1%, respectively, which were 9.8% and 11.3% higher than those of the A356 alloy. The best comprehensive tensile mechanical properties of the alloys in the as-cast state were those of the A356SCU alloy, and its UTS and EL reached 186 MPa and 12.8%, respectively; these were 25.7% and 345.9% higher than those of the A356 alloy. As shown in Figure 8c–d, the UTS values of the alloys after the T6 treatment were significantly increased compared with the as-cast state, while the EL showed no obvious change. After ultrasonic treatment, the UTS of the alloys stabilized at 285 ± 3 MPa, but the plasticity was obviously improved. In particular, the EL of A356SCU reached 12.7%, which was 8.5% higher than that of the A356SC alloy.

To investigate the fracture mechanism, fracture morphologies were observed by SEM, as shown in Figure 9. Figure 9a,d shows that the fracture surfaces of the A356 alloys without modifiers were composed of many coarse deconstruction planes that resulted from the brittle fracture mode. After the ultrasonic treatment, the numbers of tearing ribs and deconstruction planes were increased, and the size decreased, but the fracture mode had no obvious change. Figure 9b,e shows that the number of cleavage planes sharply increased and that many small dimples were evenly distributed on the fracture surface. However, there were still a few cleavage surfaces with particle sizes of 10 μm. The fracture mode changed to a mixed mode dominated by ductile fracture. After the ultrasonic treatment, the areas of the cleavage planes on the fracture surface were finer and the dimple size was increased, which resulted in an increase in ductility. Figure 9c,f shows that the fracture morphology of A356SC was similar to that of the A356S alloy, but the number of dimples on the A356SC alloy increased significantly, and the ductility improved slightly. After the ultrasonic treatment, the cleavage plane of the fracture was refined further, and the dimple sizes were increased. The bright white phase in the fracture of Figure 9g was analyzed by energy spectroscopy, and the results showed that the phase was an Al-Si-Ce ternary compound with a length of approximately 1–2 μm; this was distributed with eutectics at the dendrite interstices.

## 4. Discussion

### 4.1. Mechanism for Grain Refinement by Ultrasonic Treatment

Heterogeneous nucleation theory [29] indicates that high-temperature intermetallic compounds with crystal structures similar to that of the α-Al phase in the Al melt may be effective nucleation substrates for α-Al and favor the production of fine α-Al.

Figure 10 shows that each curve has two obvious exothermic peaks, which correspond to phase transformations of Al-Si (Peak B) and α-Al (Peak A). Sr is a common modifier used in Al-Si alloys. The addition of Sr had nearly no effect on the nucleation of α-Al [1,30] but significantly increased the size of the α-Al SDAS because the temperature of the Al-Si eutectic reaction was reduced by the addition of Sr. Therefore, the temperature range for solidification was widened, and the growth space for α-Al was increased, which led to the coarsening of the α-Al dendrites. Ce has the potential to form nucleation sites in Sr/Ce synergistic modifications. According to the mismatch model of Bramfit [31], Al11Ce3 contains nucleation sites for α-Al, as the lattice mismatch between Al11Ce3 and α-Al is 7.19%. [32]. However, the Ce content used in this study was far below the Ce content needed to form Al11Ce3, and Ce mainly existed as an Al-Si-Ce ternary phase, as shown in Figure 9h; this cannot form effective heterogeneous nucleation particles. The addition of Sr/Ce decreased the temperature of the Al-Si eutectic reaction (from 571.9 °C to 564.6 °C), as shown in Figure 10. The decreasing nucleation temperature of the eutectic Si increased the subcooling of its nucleation, which promoted the nucleation ratio of the eutectic Si.

Ultrasonic treatment has a significant refining effect in different alloy modifications, and its mechanism is mainly related to shock waves and microjets generated when ultrasonic bubbles collapse [19]. First, the shock waves and microjets generated by the ultrasound activated the impurities in the melt. Second, the high pressure generated by the shock wave changed the interface energy between the impurity and melt, improved the wetting ability and reduced the heterogeneous nucleation energy barrier. A high-speed microjet immersed the melt in the grooves and cracks of impurities, which enhanced capillary action [33]. In addition, the curvature of these cracks and grooves promoted melt activation and supercooling nucleation. Therefore, the ultrasonic treatment increased the number of heterogeneous nucleation sites and refined the α-Al grains.

### 4.2. Modification Mechanism of Eutectic Si

According to the model of impurity-induced twinning (IIT) [34], the atoms of modification elements adhere to the growth steps of Si, which prevents the conversion Si into a lamellar shape. A large number of twins formed during the growth of the Si crystals, leading to a morphological transformation of the eutectic Si into fibrous Si. Sr, a surface-active element, easily attached to the growth step of Si and hindered the conversion of Si into flakes in the step growth mechanism. In synergistic modifications caused by Sr/Ce, the strong structural supercooling resulting from the addition of Ce [1] reduced the rates for diffusion of Sr atoms from Si to Al, caused more Sr atoms to remain on the surface of Si and enhanced the modification effects of Sr.

Ultrasonic vibration activated the oxide inclusions [19], Sr and Ce atoms, and promoted the adsorption of Sr onto the eutectic Si surface, which further limited the step growth of the eutectic Si. The stepped growth of the eutectic Si particles was suppressed, and increasingly thinner branches were formed by twin growth mechanisms. These multibranched eutectic Si particles more easily fused and spheroidized during the solution treatment, leading to a smaller size of the eutectic Si.

### 4.3. Mechanism of Strengthening

The strengthening mechanism of the A356 alloy mainly includes fine grain strengthening, second phase strengthening and solution strengthening [35,36,37].

The strengthening coefficient T of the alloy can be calculated as:(1)$$T=\sigma_{A}+\sigma_{B}+\sigma_{C}$$
where $\sigma_{A}$, $\sigma_{B}$ and $\sigma_{C}$ represent second phase strengthening, fine grain strengthening and solution strengthening effects, respectively.

According to the Halle–Petch equation [38], the strength of the alloy with respect to grain size is expressed as:(2)$$\sigma_{A}=\sigma_{0}+K{\left(d\right)}^{-1/2}$$
where $\sigma_{0}$ and K are constants and (*d*) is the grain diameter. From Equation (2), the diameter (*d*) of the strength grains in polycrystals is on the (−1⁄2) order of magnitude, i.e., the finer the alloy grains, the higher the strength.

According to the Orowan mechanism [39] for the process of alloy deformation, the dislocation bypassed second particles with low deformabilities during deformation, and a dislocation ring remained around the particle, which led to the proliferation of dislocation.

The critical tangential stress of the dislocation line depends on the minimum radius of curvature ($\frac{d}{2}$) when bypassing the particle barrier. Therefore, the critical tangential stress of the dislocation line needed to pass through the particle is Δτ = $\left(\frac{T}{b}\right)\times \frac{d}{2}$, and T is the dislocation line tension. Since T≈$\frac{1}{2}$Gb^2^, b is the mode of Park’s vector, which is simplified to Δτ ≈ $\frac{Gb}{d}$, as in:(3)$$\Delta \tau :\approx \cdot \frac{Gbf^{1/2}}{d}\ln\left(\frac{2r}{r_{0}}\right)\cdot \approx \cdot \alpha f^{1/2}r^{-1}$$
where α is a constant and (*f*) is the volume fraction of particles. As the ionic radius (*r*) or particle spacing (*d*) decreases, the strengthening effect increases. The smaller the radii of eutectic silicon particles and the smaller the particle space, the higher is the strengthening of the alloy.

Mg_2_Si plays a decisive role in strengthening the A356 alloy during the T6 heat treatment [40,41], which is the reason for the significant increase in strength after the T6 heat treatment. The strength difference caused by solution strengthening can be ignored due to the similar composition and heat treatment. After the ultrasonic treatment, the strength of the as-cast alloy is obviously improved, which is mainly due to the fine grain strengthening and second phase strengthening. Second phase strengthening, especially precipitation strengthening, is dominant during the T6 heat treatment, and the strength improvement effect brought by ultrasound is not obvious, since $\sigma_{B}\gg \sigma_{A}+\sigma_{C}$.

The sharp edge of the eutectic Si in alloy A356 easily caused stress concentration [6], which promoted matrix splitting and cracking. After the ultrasonic treatment, the reduction in the eutectic Si size effectively dispersed the stress. Furthermore, the finer grains enlarged the grain boundary area and delayed crack rack propagation, thus improving the plasticity of the alloy. After Sr modification and Al-5Sr-5Ce synergistic modification, the morphology of the eutectic Si was transformed into rods [1], which effectively decreased the stress concentration. The ultrasonic treatment refined the sizes of the grains and obviously improved the plasticity of the alloy.

## 5. Conclusions

The effects of the ultrasonic treatment on the microstructure and tensile properties of the A356 alloys under different modification treatments were investigated, and the main conclusions are as follows:Ultrasonic treatment refined α-Al grains under different modification treatments. After the ultrasonic treatment, the coarse columnar and equiaxed Al grains were transformed into evenly distributed equiaxed grains with different modifiers. The SDAS of the A356 alloy with Sr modification was reduced by 8.5% compared with the alloy with no ultrasonic treatment.The eutectic Si of the unmodified A356 alloy had no obvious change by the ultrasonic treatment, but the branch diameter of the eutectic Si becomes smaller in the Sr and Sr/Ce modification alloys after the ultrasonic treatment.The UTS and EL of the as-cast alloys with different modifiers were improved by the ultrasonic treatment. In particular, the strength and elongation of the unmodified alloy were increased by 9.8% and 11.3%, respectively.After the T6 treatment, the UTS values for the alloys with different modifications showed no obvious change due to the ultrasonic treatment, but the ultrasonic treatment significantly improved the plasticity of the alloy with the Sr/Ce modification.The improvement in UTS in the as-cast alloys was caused by the refinement of α-Al, while the improvement in the plasticity in the T6 state was caused by the modification of the eutectic Si.

## Figures and Tables

**Figure 1 materials-15-03714-f001:**
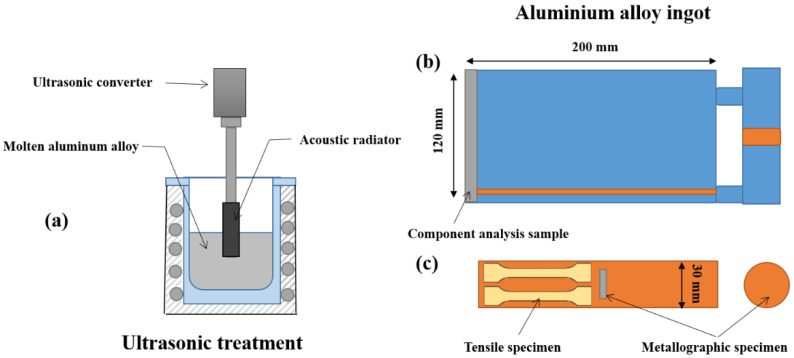
Schematic diagram of the ultrasonic casting experiment and specimen sampling location: (**a**) schematic diagram of ultrasonic treatment; (**b**) Al alloy ingot; (**c**) sample sampling position.

**Figure 2 materials-15-03714-f002:**
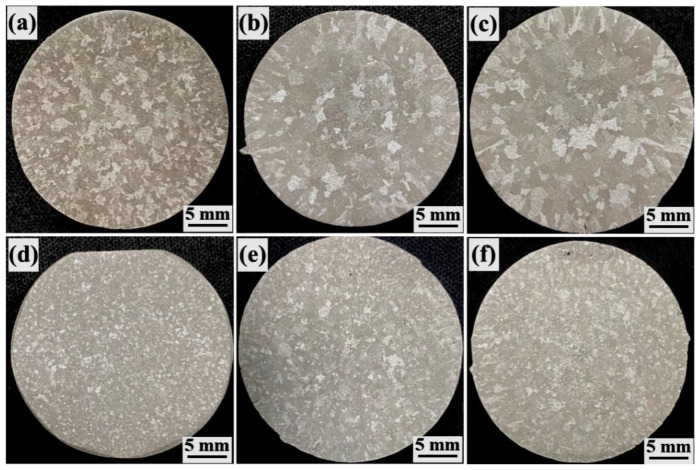
Macrostructures of A356 alloys treated under different conditions: (**a**) A356, (**b**) A356S; (**c**) A356SC; (**d**) A356U; (**e**) A356SU; and (**f**) A356SCU.

**Figure 3 materials-15-03714-f003:**
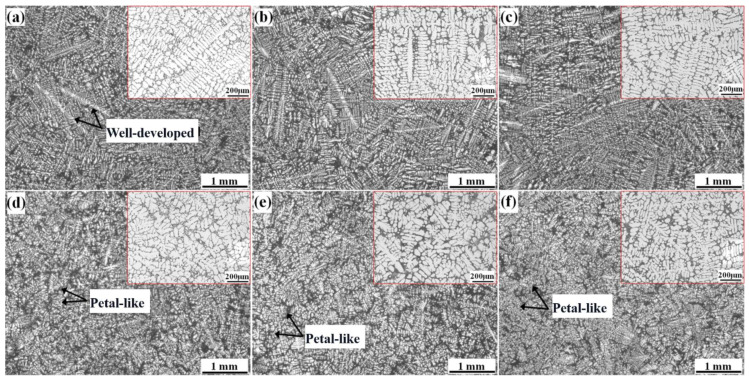
OM images of α-Al morphologies treated under different conditions: (**a**) A356; (**b**) A356S; (**c**) A356SC; (**d**) A356U; (**e**) A356SU; (**f**) A356SCU.

**Figure 4 materials-15-03714-f004:**
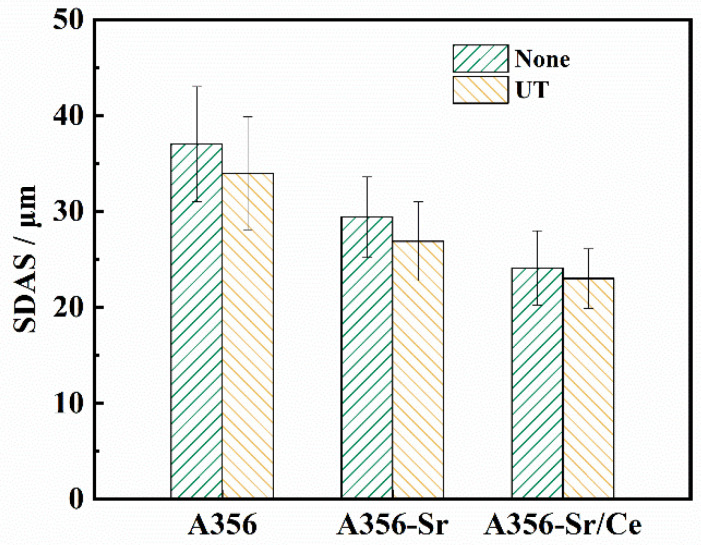
SDAS values of A356 alloy under different treatment conditions (without ultrasonic treatment (none); ultrasonic treatment (UT)).

**Figure 5 materials-15-03714-f005:**
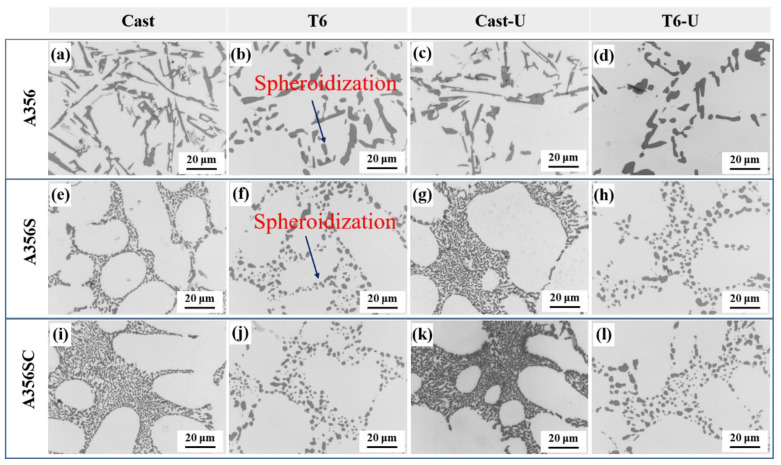
OM images of eutectic Si under different treatment conditions: (**a**) A356; (**b**) T6-A356; (**c**) A356U; (**d**) T6-A356U; (**e**) A356S; (**f**) T6-A356S; (**g**) A356SU; (**h**) T6-A356SU; (**i**) A356SC; (**j**) T6-A356SC; (**k**) A356SCU; (**l**) T6-A356SCU.

**Figure 6 materials-15-03714-f006:**
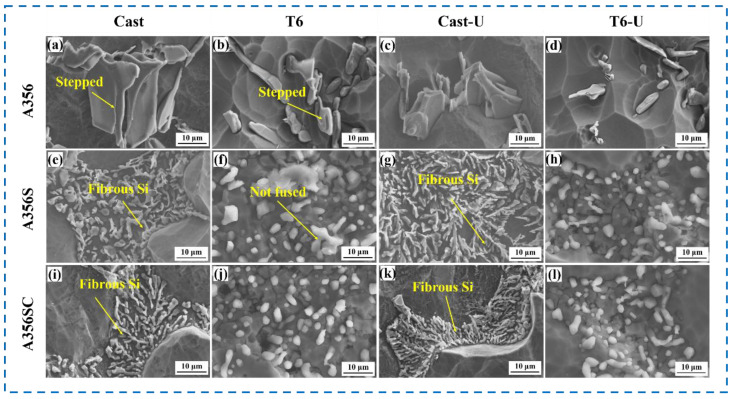
SEM images of deep-etched eutectic Si of A356 alloys under different treatment conditions: (**a**) A356; (**b**) T6-A356; (**c**) A356U; (**d**) T6-A356U; (**e**) A356S; (**f**) T6-A356S; (**g**) A356SU; (**h**) T6-A356SU; (**i**) A356SC; (**j**) T6-A356SC; (**k**) A356SCU; (**l**) T6-A356SCU.

**Figure 7 materials-15-03714-f007:**
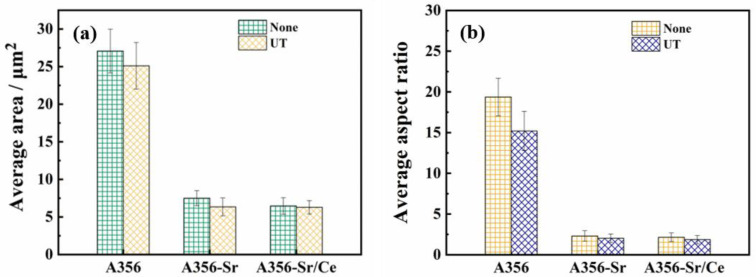
Average area (**a**) and aspect ratio (**b**) of eutectic Si in the A356 alloy under different treatment conditions.

**Figure 8 materials-15-03714-f008:**
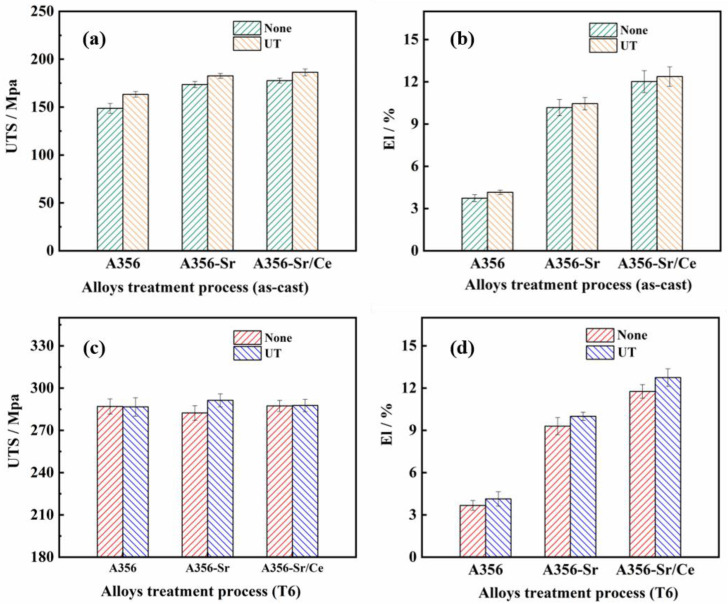
Tensile mechanical properties of A356 alloy under different treatment conditions: (**a**) ultimate tensile strength under as-cast condition; (**b**) elongation under as-cast condition; (**c**) ultimate tensile strength under the T6 condition; (**d**) elongation under the T6 condition.

**Figure 9 materials-15-03714-f009:**
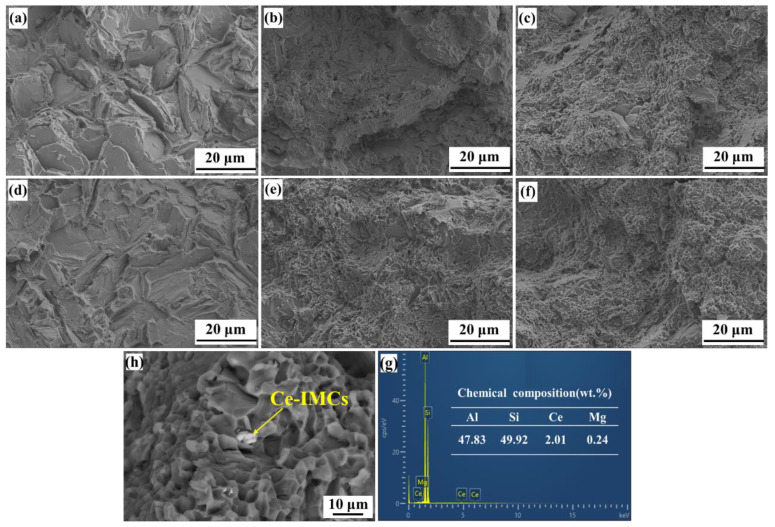
SEM images of fracture morphologies for as-cast A356 alloy under different treatment conditions and chemical composition of Ce-IMCs: (**a**) A356; (**b**) A356S; (**c**) A356Sc; (**d**) A356U; (**e**) A356SU; (**f**) A356SCU; (**h**) A356SC; (**g**) EDS result for Ce-IMCs.

**Figure 10 materials-15-03714-f010:**
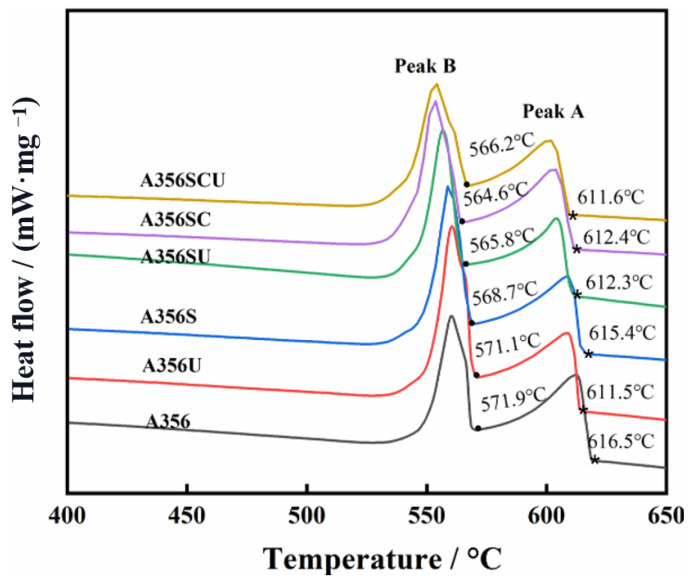
DSC curves of A356 alloys after different modifications and melt treatments.

**Table 1 materials-15-03714-t001:** Chemical compositions of the experimental alloys (wt.%).

Melt Treatment Process	No.	Si	Fe	Mg	Sr	Ce	Al
A356 (Base alloy)	A356	6.89	0.12	0.38	-	-	Bal.
A356-Ultraphonic	A356U	6.85	0.11	0.39	-	-	Bal.
A356-Sr	A356S	7.15	0.10	0.37	0.013	-	Bal.
A356-Sr-Ultraphonic	A356SU	7.03	0.10	0.41	0.012	-	Bal.
A356-Sr/Ce	A356SC	7.03	0.12	0.41	0.014	0.045	Bal.
A356-Sr/Ce- Ultraphonic	A356SCU	6.78	0.09	0.37	0.011	0.041	Bal.

## Data Availability

The datasets generated and analyzed during the current study are available from the corresponding author on reasonable request.

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
