# Peer review of "Effect of Ultrasonic-Assisted Modification Treatment on the Microstructure and Properties of A356 Alloy"

_materials, 2022, doi:10.3390/ma15103714_

Round 1
Reviewer 1 Report
Hu et al. prepared unmodified, Sr-modified and Sr/Ce-modified A356 casting alloys with/without ultrasonic melt treatment and with/without T6 heat treatment. They compared the alloys’ microstructure (primary Al and eutectic Si phase) using optical and scanning electron microscopy and measured the alloys’ mechanical properties (tensile strength, elongation).
A macroscopic refinement of primary Al phase was found in the ultrasonic-treated alloys with respect to samples without ultrasonic treatment (Fig. 2). However, on a microscopic scale, the refinement seems less significant. The petal-like shape after ultrasonic treatment, as seen by the authors (Fig. 3b, d, f), cannot be confirmed to be representative. The authors report a slight decrease of the SDAS values due to ultrasonic treatment (4.2-8.5 %, see Fig. 4). The SDAS values of unmodified and Sr/Ce-modified A356 alloys (~34 and 24 µm, respectively) differ substantially from previously reported values [Wu et al. J. Alloys Compd. 791, 2019], where the addition of Sr/Ce without ultrasonic treatment reduced the SDAS from 59 (unmodified) to 15 µm (Sr/Ce-modified). The reason for this difference should be commented. Furthermore, details about quantitative image analysis (number of analyzed images, total analyzed area) are not given, leaving some doubts if the present quantification is indeed statistically significant.
Concerning the influence of ultrasonic treatment on the eutectic Si morphology, the interpretation of the microscopic images in Fig. 5 and 6 seems vague. Some of the statements (e.g., increased branches of eutectic Si by ultrasonic treatment) cannot be confirmed without a true 3D characterization (e.g., via focused ion beam/SEM tomography). The Si phase before ultrasonic treatment might be as interconnected as after ultrasonic treament, but the connection sites simply lies above or below the cross-sectional cuts. It only becomes clear from Figs. 5 and 6 that after the T6 heat treatment, the eutectic Si appears more spherical.
Furthermore, the authors cite only very recent literature about eutectic modification (mainly from 2019-2021), which does not reflect the vast number of studies published in this field. The discussion section of the manuscript goes beyond the results presented and should be carefully revised.
Based on the above considerations, I cannot recommend the manuscript for publication in Materials.
Author Response
Response to Reviewers
Thank you for your letter and for the reviewers’ comments concerning our manuscript entitled “Effect of Ultrasonic-assisted Modification Treatment on the Microstructure and Properties of A356 Alloy”. Those comments are all valuable and very helpful for revising and improving of this paper. We have studied comments carefully and made corrections which we hope meet with approval. All the revised portions are marked in red colour.
Reviewer 1#
1) A macroscopic refinement of primary Al phase was found in the ultrasonic-treated alloys with respect to samples without ultrasonic treatment (Fig. 2). However, on a microscopic scale, the refinement seems less significant. The petal-like shape after ultrasonic treatment, as seen by the authors (Fig. 3b, d, f), cannot be confirmed to be representative.
Response: Thank you very much for your comments. In order to reflect the effect of ultrasonic treatment on the morphology and size of α-Al dendrites, the lower magnifications (25×) OM images were supplied, instead of 50× OM images. Under a larger field of view, the well-developed α-Al dendrites were observed before the ultrasonic treatment (Fig. 3a, b, c). After ultrasonic treatment, the length of dendrites were significantly reduced, and a part of dendrites were changed into petal-like (Figs. 3d, e, f).
2)The authors report a slight decrease of the SDAS values due to ultrasonic treatment (4.2-8.5 %, see Fig. 4). The SDAS values of unmodified and Sr/Ce-modified A356 alloys (~34 and 24 µm, respectively) differ substantially from previously reported values [Wu et al. J. Alloys Compd. 791, 2019], where the addition of Sr/Ce without ultrasonic treatment reduced the SDAS from 59 (unmodified) to 15 µm (Sr/Ce-modified). The reason for this difference should be commented.
Response: as well known, the cooling rate has an important effect on the SDAS of the alloy. For Example, Liu et al. [Effects of cooling rate on the microstructure and tensile strength of A356 alloy wheels, 3rd International Conference on Material, Mechanical and Manufacturing Engineering (IC3ME 2015). Atlantis Press, (2015) 2103-2109] reported that the SDAS reduced from 44.41 µm to 31.86 µm as the cooling rate increased from 0.69 ℃/s to 1.95 ℃/s. Careful study of experimental details of Wu [Wu et al. J. Alloys Compd. 791, 2019], the shape, geometry and preheated temperature of metal mold existed a significant difference from our work, which affect the cooling rate of the alloy. In addition, the alloy composition also has an important effect on the grain size of the alloy. In particular elements thought to have refined grains such as RE. The highest Ce content (0.081 wt.%) in Wu’s work is high than that of our work. Therefore, the SDAS differences of Wu and our work can be summed up as two factors: the shape, geometry and preheated temperature of metal mold, and the Ce content.
3) Furthermore, details about quantitative image analysis (number of analyzed images, total analyzed area) are not given, leaving some doubts if the present quantification is indeed statistically significant.
Response: in this work, the statistical data of α-Al and eutectic Si is analyzed by Image Pro-Plus 6.0 software. For the SDAS, we selected about 30 OM images of 50 times. The area of a single image is approx. 3 mm2, and the total analysis area is about 90 mm2. For the statistical data of eutectic Si, the we selected about 50 OM images of 1000 times. The area of a single image is approx. 0.3 mm2, and the total analysis area is about 15 mm2. The details about quantitative image analysis has been supplied in revised manuscript.
4) Concerning the influence of ultrasonic treatment on the eutectic Si morphology, the interpretation of the microscopic images in Fig. 5 and 6 seems vague. Some of the statements (e.g., increased branches of eutectic Si by ultrasonic treatment) cannot be confirmed without a true 3D characterization (e.g., via focused ion beam/SEM tomography). The Si phase before ultrasonic treatment might be as interconnected as after ultrasonic treatment, but the connection sites simply lies above or below the cross-sectional cuts. It only becomes clear from Figs. 5 and 6 that after the T6 heat treatment, the eutectic Si appears more spherical.
Response: we changed the “three-dimensional morphology of eutectic Si” to “the morphology of deep-etched eutectic Si”. The description in the manuscript was also corrected.
5)Furthermore, the authors cite only very recent literature about eutectic modification (mainly from 2019-2021), which does not reflect the vast number of studies published in this field. The discussion section of the manuscript goes beyond the results presented and should be carefully revised.
Response: Thank you very much for your suggestion. We have supplied relevant literature of eutectic Si modification. Then the discussion section of the manuscript has been carefully revised.
We tried our best to improve the manuscript and made some changes in the manuscript. These changes will not influence the content and framework of the paper.
We appreciate for Editors/Reviewers’ warm work earnestly and hope that the correction will meet with approval.
Once again, thank you very much for your comments and suggestions.
Yours Sincerely
Dongfu Song

Reviewer 2 Report
The paper is clearly and carefully writen.
Author Response
Thank you very much.
Reviewer 3 Report
Dear authors,
thank you for your interesting contribution of Ultrasonic-assisted treatment of A356. You present an explorative study which includes modified alloy variants, which is of great interest for casting processes.
I have the following comments:
1. Sample preparation:
- You did not mention, under which atmosphere the samples were prepared. I assume this is done in an inert atmosphere?
- If I understood your process right, the melt is treated with ultrasonic, but ultrasonic is not applied during solidification? What is the hypotheses for this treatment?
- how long was the solidification process and the total cooling time until room temperature? Was the solidification rate of each experiment the same? Please give cooling rates. If cooling rates are not known, a comparison of SDAS does not make sense.
- was the feedstock mass in alle experiments the same? This will also affect the cooling rate
- please add dimensions of each sample if the difference is significant (otherwise give with standard deviation)
- what is the reproducibility of your experiments? Can you repeat each experiment and get the same microstructure?
2. Results
- Where were the samples cut for micrographs? I assume there is a pronounced axial gradient, but you are showing horizontal slices in Fig. 2. It may be more meaningful to show vertical cross-sections to see the influence on the growth
- In which direction were the images in Fig. 3 taken? How did you measure the SDAS, how many arms, how many dendrites for each data point? What is the difference of SDAS at different spots in the sample?
- mechanical properties: the difference between „none“ and „UT“ is within the error bar most of the time. As long as the cooling rate of each sample is not known, this is not significant.
3. Discussion
- You state that the shift of the melting range by 5 degC has a major influence on the cooling. Fig. 10 shows that the solidification range is not widened, but only shifted. Such small changes of composition will also not affect latent heat. In terms of heat transfer with solidification, such a shift is not significant for the solidification rate during solidification, where the dendrites are formed. Since both temperatures are shifted, driving temperature gradients will be maintained.
The assumption that a decrease of 571.9 to 564.6 degC leads to a longer solidification time is wrong.
- have you checked the reproducibility of DSC with different samples from each ingot? You easily get a shift of a few degrees.
In summary, I have to conclude that the manuscript is not appropriate for publication in its current state. There are many flaws with the experimental procedure of sample preparation which forms the base for all following results. Process conditions during casting are missing, so that the samples can not be compared.
Author Response
Response to Reviewers
Thank you for your letter and for the reviewers’ comments concerning our manuscript entitled “Effect of Ultrasonic-assisted Modification Treatment on the Microstructure and Properties of A356 Alloy”. Those comments are all valuable and very helpful for revising and improving of this paper. We have studied comments carefully and made corrections which we hope meet with approval. All the revised portions are marked in red colour.
Reviewer #3:
- Sample preparation:
- You did not mention, under which atmosphere the samples were prepared. I assume this is done in an inert atmosphere?
Response: the ultrasonic treatment process was carried out under the Ar atmosphere, which supplied in revised manuscript.
- If I understood your process right, the melt is treated with ultrasonic, but ultrasonic is not applied during solidification? What is the hypotheses for this treatment?
Response: the ultrasonic treatment was applied on the melt before solidification. The hypotheses of ultrasonic treatment was mainly reflected in two aspects. First, the ultrasonic treatment can homogenize the melt composition and break up the residual oxide film in the melt. Second, the ultrasonic treatment was beneficial to the activation of the oxide film, and improved the nucleation rate of the α-Al. The relevant theoretical basis can refer to the work of Balasubramani [ N. Balasubramani, D. StJohn, M. Dargusch, et al. Ultrasonic Processing for Structure Refinement: An Overview of Mechanisms and Application of the Interdependence Theory[J]. Materials, 2019, 12(19): 3187.]
- how long was the solidification process and the total cooling time until room temperature? Was the solidification rate of each experiment the same? Please give cooling rates. If cooling rates are not known, a comparison of SDAS does not make sense.
Response: the cooling time of solidification process was ~70 s. To maintain a relatively constant of cooling rate and solidification time, we have taken several measures to keep the same solidification rate of each experiment. First, the temperature of melt was holding at 720 °C before pouring; second, the metal mold and its preheated temperature was strictly controlled. In this work, the cooling rate was about 2~3 °C /s from previous experience. Under the relatively consistent condition of melt temperature, pouring process, the mold and its preheating temperature, the SDAS of the alloys with different melt treatments still has a certain reference sense.
- Was the feedstock mass in alle experiments the same? This will also affect the cooling rate. Please add dimensions of each sample if the difference is significant (otherwise give with standard deviation)
Response: yes, the raw materials mass of each alloy is about 2 kg. During the processes of smelting, melt treatment and pouring, there was an inevitably loss mass of the ingot, but the loss ratio was less than 0.5%. Namely, the ingots’ mass was kept at 1930±30g.
-What is the reproducibility of your experiments? Can you repeat each experiment and get the same microstructure?
Response: due to the relatively simple experiments process and more than 8 years experimental experience, we are very confident in the reproducibility of our experiments. In this paper, the preparation process of each alloy is similar, the factors affecting the alloy microstructure are the modification process and ultrasonic treatment. Therefore, we can repeat each experiment and get the same microstructure.
- Results
- Where were the samples cut for micrographs? I assume there is a pronounced axial gradient, but you are showing horizontal slices in Fig. 2. It may be more meaningful to show vertical cross-sections to see the influence on the growth
Response: Thank you very much for your suggestion. We have re-taken the cross-sectional specimens and observed their microstructure. The results (new Fig.3 for lower multiple) obtained from vertical cross-sections have not exhibited a significant gradient change.
- In which direction were the images in Fig. 3 taken? How did you measure the SDAS, how many arms, how many dendrites for each data point? What is the difference of SDAS at different spots in the sample?
Response: Response: As shown in Fig.1b and c, the OM sample position of each ingot was the same. In this work, the statistical data of α-Al and eutectic Si is analyzed by Image Pro-Plus 6.0 software. For the SDAS, we selected about 30 OM images of 50 times. The area of a single image is approx. 3 mm2, and the total analysis area is about 90 mm2. The arms and dendrites for each data point were 75 and 750, respectively. As shown in Figure 2, the structure difference between the center and edges of the sample was obvious. Therefore, we have taken three fields of view along the thickness direction of the OM sample. There is a certain difference in the SDAS of each view, and the data point mean the average SDAS value of the three fields.
- mechanical properties: the difference between „none“ and „UT“ is within the error bar most of the time. As long as the cooling rate of each sample is not known, this is not significant.
Response: the strengthening mechanism of the A356 alloy mainly includes fine grain strengthening, second phase strengthening and solution strengthening. After ultrasonic treatment, the grain size in the alloy reduced to some extent had been refined to a certain extent, but the properties improvement by the grain refinement had a certain limit. Furthermore, the range of the ordinate (Y-axis) was too large on the Fig. 8, which displayed a small difference in visual.
- Discussion
- You state that the shift of the melting range by 5 deg C has a major influence on the cooling. Fig. 10 shows that the solidification range is not widened, but only shifted. Such small changes of composition will also not affect latent heat. In terms of heat transfer with solidification, such a shift is not significant for the solidification rate during solidification, where the dendrites are formed. Since both temperatures are shifted, driving temperature gradients will be maintained.
-The assumption that a decrease of 571.9 to 564.6 degC leads to a longer solidification time is wrong.
Response: we appreciate your detailed comments and specific criticisms. We have corrected the description of the eutectic Si nucleation temperature reduction for DSC. The decreasing nucleation temperature of eutectic Si increased the subcooling of its nucleation, which helped to increase the nucleation of eutectic Si.
- have you checked the reproducibility of DSC with different samples from each ingot? You easily get a shift of a few degrees.
Response: we did not repeat the DSC tests on each ingot, but we have been repeatedly tested the chemical composition of each ingot. The results showed that the compositions at each longitudinal height did not show significant differences. As we known, the temperature reduction of Al-Si eutectic brought by Sr modification is only a few degrees, so we have reason to believe that the temperature reduction in this work was caused by ultrasonic treatment.
We tried our best to improve the manuscript and made some changes in the manuscript. These changes will not influence the content and framework of the paper.
We appreciate for Editors/Reviewers’ warm work earnestly and hope that the correction will meet with approval.
Once again, thank you very much for your comments and suggestions.
Dongfu song
